# Giant thermal expansion of a two-dimensional supramolecular network triggered by alkyl chain motion

Sebastian Scherb [1,4], Antoine Hinaut [1,4]*, Rémy Pawlak [1], J.G. Vilhena [1]*, Yi Liu[2], Sara Freund [1], Zhao Liu[1], Xinliang Feng [3], Klaus Müllen[2], Thilo Glatzel [1], Akimitsu Narita [2] & Ernst Meyer [1]*

Thermal expansion, the response in shape, area or volume of a solid with heat, is usually large in molecular materials compared to their inorganic counterparts. Resulting from the intrinsic molecule flexibility, conformational changes or variable intermolecular interactions, the exact interplay between these mechanisms is however poorly understood down to the molecular level. Here, we investigate the structural variations of a two-dimensional supramolecular network on Au(111) consisting of shape persistent polyphenylene molecules equipped with peripheral dodecyl chains. By comparing high-resolution scanning probe microscopy and molecular dynamics simulations obtained at 5 and 300 K, we determine the thermal expansion coefficient of the assembly of $980 \pm 110 \times 10^{-6} \, K^{-1}$, twice larger than other molecular systems hitherto reported in the literature, and two orders of magnitude larger than conventional materials. This giant positive expansion originates from the increased mobility of the dodecyl chains with temperature that determine the intermolecular interactions and the network spacing.

[1] Department of Physics, University of Basel, Klingelbergstrasse 82, 4056 Basel, Switzerland. [2] Max Plank Institute for Polymer Research, Ackermannweg 10, 55128 Mainz, Germany. [3] Faculty of Chemistry and Food Chemistry, TU Dresden, Mommsenstrasse 4, 01069 Dresden, Germany. [4] These authors contributed equally: Sebastian Scherb, Antoine Hinaut *email: antoine.hinaut@unibas.ch; guilhermevilhena@gmail.com; ernst.meyer@unibas.ch

The expansion of a material upon heating, known as thermal expansion is a fundamental property of solid systems that governs many of their mechanical applications[1–3]. Typically, thermal expansion coefficients ($\alpha = \frac{\Delta l}{l_0 \Delta T}$ with $\Delta l$ is the length variation, $l_0$ the initial length and $\Delta T$ the change of temperature) of inorganic materials are in the range of $0-20 \times 10^{-6}\,\mathrm{K}^{-1}$, whereas molecular counterparts vary from $50-60 \times 10^{-6}\,\mathrm{K}^{-1}$ (poly(styrene))[2] up to $\sim\!420 \times 10^{-6}\,\mathrm{K}^{-1}$ for porous coordination polymers[1]. Compared to inorganic materials, molecular architectures enable larger positive (or even negative[4,5]) expansions due to various mechanisms involving the intrinsic molecules flexibility, their propensity to conformational changes or the weak intermolecular interactions[2]. To date, however, understanding of the underlying mechanisms governing such large expansion coefficients at the molecular scale remains elusive.

Supramolecular chemistry has opened these last years new avenues in controlling spontaneous bottom-up assemblies of molecules both in solution[6,7], in the solid state or at the solid interfaces[8–11]. Resulting from weak non-covalent intermolecular interactions, supramolecular assemblies enable the growth of well-extended and highly-ordered structures with tunable chemical, optochemical or mechanical properties[9–13]. In aqueous environments, extended polycyclic hydrocarbon molecule (PAH) assemblies have been also reported through the interdigitation of long alkyl chains[14–18]. While these organic architectures have already shown potential applications in photovoltaics and nanoelectronics[19], the weak intermolecular interactions governing these assemblies and the intrinsic flexibility of the precursors also offer new opportunities for the design of responsive materials to external stimuli[6,20]. For instance, the thermal response of 2D supramolecular assemblies has never been investigated down to the molecular scale.

To address supramolecular systems at the nanoscale, atomic force microscopy (AFM) and scanning tunneling microscopy (STM) are suitable techniques since they allow the highest spatial resolutions of single molecules and their assemblies at surfaces[21–25]. The growth of two-dimensional supramolecular networks is nowadays successfully achieved at surfaces by the thermal evaporation of precursors in ultra high vacuum (UHV)[8–10,23,25,26]. This approach, however, limits the size of the precursors and prevents the use of fragile moieties such as long alkyl chains that might degrade during evaporation. In liquid, such size or composition limitations are not present[11,13–18]. In UHV conditions, the electrospray deposition (ESD) technique allows the introduction of large molecules. Therefore, it is possible to study in UHV conditions the adsorption properties of molecules and their assemblies that mimic established assemblies at the liquid–solid interface[24,27–33]. In this context, temperature-dependent STM/AFM investigations of supramolecular

systems might enable the analysis of their thermoresponse down to the atomic scale.

Here, we investigate the thermal response of a 2D supramolecular assembly composed of shape persistent spoked wheel molecules equipped with peripheral dodecyl chains using non-contact AFM (nc-AFM), STM and molecular dynamic simulations (MD) at different temperatures. By comparing atomic-scale images of the assembly at 5 and 300 K we determined a giant positive thermal expansion of about $980 \pm 110 \times 10^{-6}\,\mathrm{K}^{-1}$, which is supported by microsecond long temperature-dependent MD simulations. The alkyl chains attached to the molecular cores are found to undergo large thermal fluctuations (compared to the core) as a result of entropic effects and large anharmonic vibrations. This results in temperature-dependent intermolecular interactions that promote the giant expansion of the highly-ordered supramolecular network with temperature.

## Results

**Single spoked wheel molecules on Au(111).** The polyphenylene spoked wheel molecule (SW)[17] (Fig. 1a) is a purely sp²-hybridized molecule with a six-fold symmetry, composed of 37 benzene rings forming spokes (orange) and sides (green) of the molecular skeleton. Each spoke axis is functionalized with a peripheral dodecyl side chain.

To safely deposit the SW precursors onto Au(111), we used an electrospray deposition (ESD) device attached to the UHV setup[24,27–33]. The spray-deposition was always performed on the Au(111) substrates kept at room temperature and followed by an annealing step at 450 K. As shown in large-scale STM measurements (Supplementary Fig. 1), single SW molecules are observed intact on the surface indicating the low diffusion of the molecule on the gold surface and the absence of any degradation. Figure 1b shows a close-up STM image acquired at 5 K with a CO-terminated tip of such an isolated SW molecule. While both the core and the dodecyl chains are clearly visible, the arms (white arrow) appear fuzzier than the core due to their relative mobility under the influence of the scanning tip. Within the core, the spokes and sides regions are observed with slight differences in contrast suggesting a corrugation of the core. Contrarily to the chemical structure representation shown in Fig. 1a, the SW core adopts a three-fold symmetry.

The corresponding constant-height AFM image with a CO-terminated tip (Fig. 1c) further confirms this particular core symmetry. While each spoke parts of the core are resolved, only half of the side parts are visible in the image. The peripheral dodecyl chains are absent in the AFM image (but visible by STM). To rationalize this, we performed a 10 ns long molecular dynamic (MD) simulation of the SW molecule adsorption at 300 K

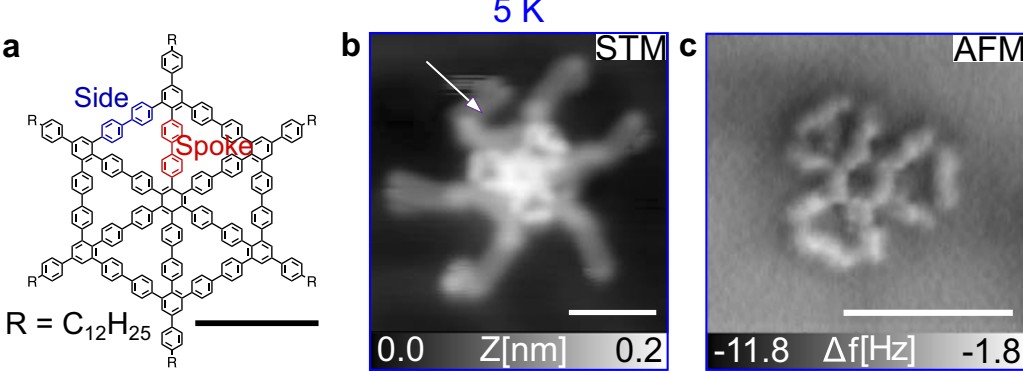

**Fig. 1 Single spoked wheel molecules on Au(111). a** Chemical structure of SW molecule equipped with dodecyl chains. **b** STM image of a single SW molecule on Au(111) ($I = 1\,\mathrm{pA}$, $U_{\mathrm{tip}} = -380\,\mathrm{mV}$). **c** Constant-height AFM image of the SW core acquired with a CO-terminated tip at 5 K. Scale bars: 2 nm.

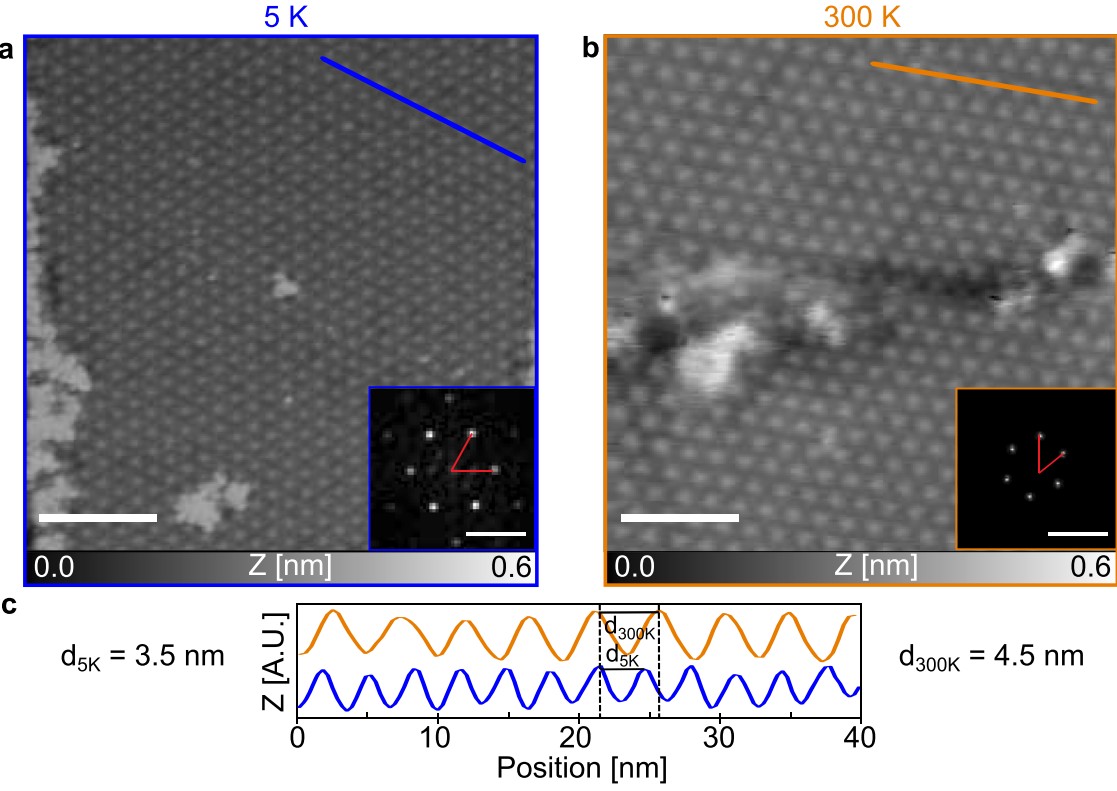

**Fig. 2 Spoked wheel molecule assembly on Au(111). a** Large scale STM topographic image of the assembly on Au(111) at 5 K. ($I = 1$ pA, $U_{tip} = -300$ mV) **b** Large scale nc-AFM image of the assembly at 300 K. ($f_2 = 1.043$ MHz, $A_2 = 0.4$ nm, $\Delta f_2 = -19$ Hz). **c** Line profiles acquired along the blue and orange lines in **a**, **b** showing a lattice parameter of 3.5 ± 0.1 nm and 4.5 ± 0.2 nm, respectively. The insets are fast Fourier transform (FFT) of the images. Scale bars: topographies: 20 nm; FFT: 0.5 nm$^{-1}$.

(Supplementary Notes 1 and 2). A large corrugation of the final conformation is observed with the dodecyl chains stacked on the Au(111) compared to the protruding core. Experimentally, these local height variations prevent constant-height AFM imaging, more sensitive to height variations compared to STM (probing a convolution of local density of states and topography), resolving the entire molecule structure.

**Thermal response of the spoked wheel assembly**. Upon spray-deposition of the spoked-wheel molecules in the sub-monolayer range, we observed different aggregates on the surface varying from single molecules up to extended assemblies (see Supplementary Figs. 1b and 2). To investigate the effect of temperature on the SW assembly, extended close-packed networks (at least 200 nm in width and length) chosen to be of similar size to avoid influences of the domain size on the lattice parameter were investigated. Figure 2a, b show STM and AFM images of the networks acquired at 5 K and 300 K, respectively. Within the assembly, individual SW cores appear as bright protrusions at both temperatures forming a long-range hexagonal network.

A different molecule spacing of the networks is measured in Fig. 2a, b as shown by the line profiles (Fig. 2c) taken along the symmetry axis and the Fast Fourier Transform of the images (see inset Fig. 2a, b). We measured lattice parameters of 3.5 ± 0.1 nm of the assembly at 5 K increasing to 4.5 ± 0.2 nm at 300 K. This corresponds to lattice shrinking of ~22% upon cooling. Note also that the preservation of the hexagonal nature of the assembly between these temperatures indicates the absence of any structural phase transitions. In conclusion, a thermal expansion coefficient of the SW on gold of $\alpha = 980 \pm 110 \times 10^{-6}$ K$^{-1}$ is

determined, which is two orders of magnitude larger than the expansion of the Au(111) of about $\alpha = 14 \times 10^{-6}$ K$^{-1}$.

**Temperature-dependent core and chain mobility**. Thermal evolution and mobility of a single SW molecule was numerically investigated using µs long all atom MD simulation on an Au(111) surface for different temperatures. Figure 3 provides a comparison of the mobility of the core and the chains of the SW molecule (see Supplementary Note 3). Snapshots of the final configuration are shown in Fig. 3a for each simulated temperature. Although displacements of the molecule core are marginal as a function of the temperature, the dodecyl chain fluctuations increase upon heating up the system (Supplementary Note 3 and Supplementary Movies 3–5). This is visualized in the snapshots (Fig. 3a) by the more straight positioning of the chains at 5 K in comparison to the increasingly tilted chains for 300 K, 420 K and finally 450 K.

To better quantify the fluctuations of the core and the chains, we compare in Fig. 3 the standard deviation of the mean square displacements (MSD) of the centre of mass of the molecule core $\delta_{core}$ and of the last atom of a dodecyl chain $\delta_{chain}$ at 5 K, 300 K, 420 K and 450 K, respectively. A larger $\delta$ corresponds to a larger mobility at a given temperature (see details in Supplementary Note 3). The core mobility is almost negligible at all considered temperatures (Fig. 3b). In contrast, the chain exhibits a much larger lateral displacement with increasing temperature. These chain fluctuations are characterized by rotations about the connection point to the core as well as in-plane bending of the chain (Supplementary Note 3 and Supplementary Movies 3–5). During simulations, the chains are continuously adsorbed on gold which is consistent with the experimental observations. The higher chain mobility compared to the core is also consistent with

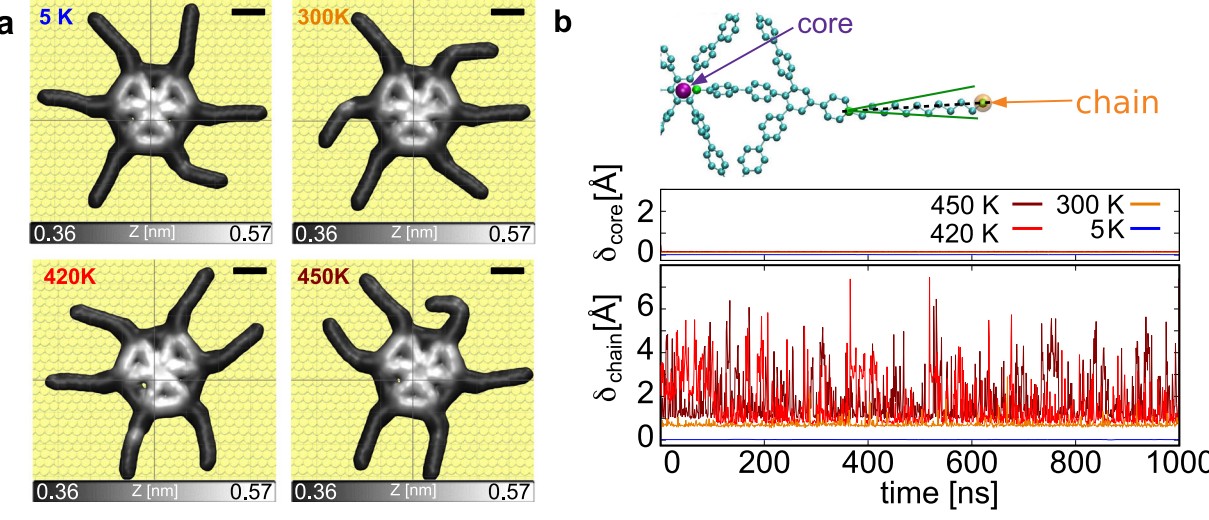

**Fig. 3 Temperature dependence of the SW core versus chains based on MD simulations. a** Final positions and configurations of single SW molecules after 1 μs at different temperatures. **b** Schematic illustration of representative positions of the molecules (top). Standard deviation of the mean square displacement (MSD) - $\delta_{core}$ - of the SW centre of mass as a function of time for different temperatures (middle). Standard deviation of the MSD - $\delta_{chain}$ - of a SW chain as a function of time for different temperatures (bottom). The MSD itself (shown in Supplementary Fig. 6) and its standard deviation are obtained through a running time average of 1 ns.

the experimental observation of accidental chain manipulations (Fig. 1b).

**Influence of interdigitated dodecyl chains on thermoresponse.** The thermal response of SW assemblies upon cooling (Fig. 4a) is numerically obtained by comparing MD simulations of a SW trimer at different temperatures. In the interest of the computational feasibility, given the sheer size and required simulation time of large assemblies, we considered a trimer whose core-core distance matched an equivalent assembly observed at 5 K (see Supplementary Fig. 4a). Further details on the simulation protocol are provided in Supplementary Note 1. Interestingly, small molecular assemblies displayed a core-core distance slightly bigger than larger assemblies, an effect possibly resulting from domain size. The simulations of this assembly (Fig. 4a, b, Supplementary Movies 1–2) confirm the significant variation of intermolecular spacing with temperature, namely from 3.9 nm at 5 K up to 5.1 nm at 450 K. The associated thermal expansion coefficient is thus $600 \pm 120 \times 10^{-6} \, \text{K}^{-1}$ in agreement with the experimental value.

Additionally, MD simulations show that the intermolecular spacing varies with the preservation of the hexagonal supramolecular structure, thereby excluding phase transition as a possible explanation of the thermal expansion. At 450 K, the chain mobility, initially constrained by the packing, increases as the network expands. This is shown by $\delta_{chains}$ as a function of time (Fig. 4c) that approaches an equilibrium value below the single molecule value on Au(111) at 450 K ($\delta_{chains}^{1SPW} = 3.34$ Å, see Supplementary Fig. 4) due to reduction by the intermolecular interaction in the assembly. While, at the single-molecule level (Fig. 3), the diffusion of the molecular core was found negligible for the considered temperatures, it appears relatively large within the assembly (Fig. 4a–c). The increased core mobility in the assembly is a consequence of the increased chain mobility at higher temperature. The mobility is enabled by the adjacent molecules forcing the network to expand.

Additionally, the expansion occurs in distinguishable events as marked by dotted lines in Fig. 4c, d. These events are separated by long periods where the core to core distance remains stable. A

lively picture of the expansion mechanism is provided in the two insets in Fig. 4c. Prior to the expansion event 2 at $t = 275$ ns, one dodecyl arm (green) is close to the neighbouring molecule (pink), that restricts its lateral fluctuation but also increases the intermolecular interactions. After the expansion event (>300 ns), alkyl chains possess enough space to adopt a more extended configuration. The relationship between chain mobility and lattice expansion is particularly visible between 400 and 700 ns in Fig. 4c, d, where the chain mobility gradually decreases prior to the third and last expansion event.

Figure 4d shows the total energy of the SW trimer as a function of time during the MD simulation at 450 K. The network expansion leads to a decrease of the internal energy of the system which is accompanied by the decrease of the number of interdigitated chains. While at low temperature, this interdigitation governs the molecular assembly to a close-packed structure and increases the internal energy, the entropic contribution to the free energy at higher temperatures, manifested by the fluctuations of the dodecyl chains, compensates the loss of internal energy of the system thus favouring the lattice expansion.

The high lattice expansion may be understood as a conjunction of three properties of the dodecyl chains: 1- a weak interaction with the surface (facilitating a rapid increase of their oscillations with temperature), 2- the soft in-plane bending stiffness of the chains (favouring this oscillation mode over other rigid vibrations) and 3- the two-dimensional constraint of the chain mobility imposed by the surface and excluding chain fluctuations out of the surface plane.

Although the finite size of our simulations might affect the chain fluctuations and the intermolecular interaction, our simulations clearly support an abnormal thermal expansion coefficient for alkyl decorated molecules. Interestingly, small molecular assemblies obtained experimentally show a larger core-core distance (see Supplementary Fig. 4a). This could possibly explain the smaller thermal expansion coefficient obtained in our simulations. At last, the large thermal expansion coefficient obtained in the experiments seems to support the idea that although in smaller assemblies the cores possess an initial higher mobility, in the larger assemblies the long experimental time scale allows enough time for the chain fluctuations equilibrate giving

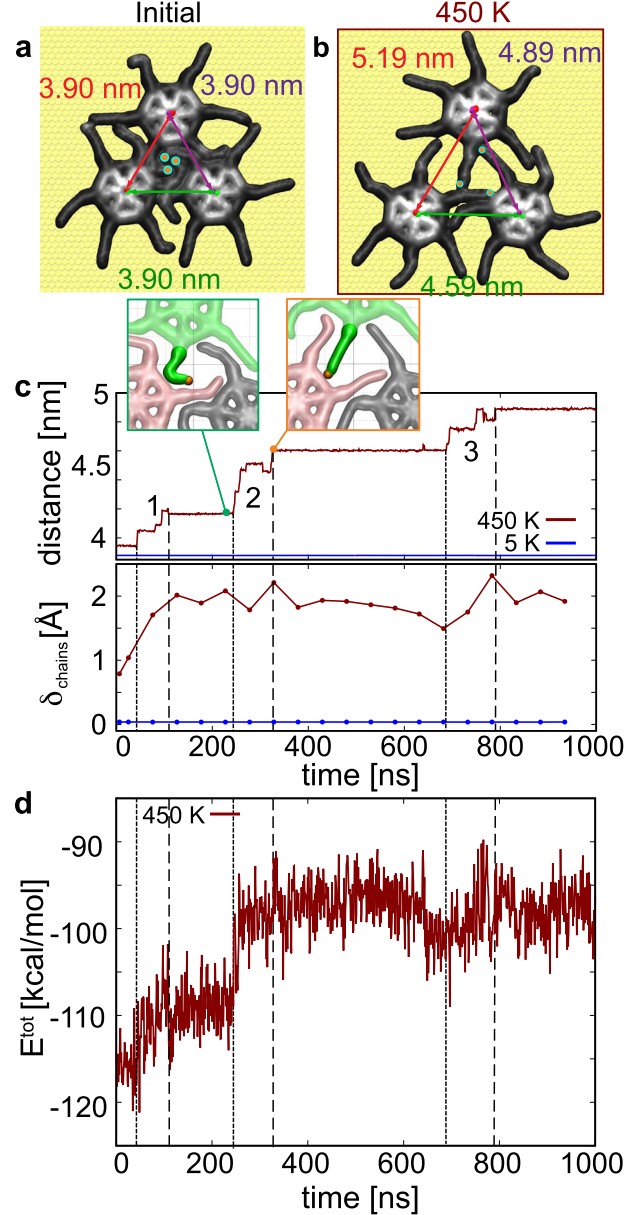

**Table 1 Thermal expansion coefficients $\alpha$ of interdigitated 2D molecular systems.**

|  | $a_{5K}$ (nm) | $a_{300K}$ (nm) | $\alpha$ ($10^{-6}$ K$^{-1}$) |
|---|---|---|---|
| SW | 3.3 | 4.5 | 980 ± 110 |
| Calculated SW | 3.9 | 4.9 (T = 450 K) | 600 ± 120 |
| HBC-6C$_{12}$H$_{25}$ from Ref. [24] | 2.2 | 2.6 | 520 ± 60 |
| HBC from ref. [34] | 1.395 (T = 1.2 K) | 1.401 | 14 |

Comparison of lattice parameters $a$ measured at 5 K and 300 K and the corresponding thermal expansion $\alpha$ for supramolecular assemblies of spoked wheel molecules (SW), pristine hexabenzocoronene molecules (HBC) and hexabenzocoronene molecules functionalized with alkyl chains (HBC-6C$_{12}$H$_{25}$). SW, HBC-6C$_{12}$H$_{25}$ and HBC are measured on large domains, SW calculated on a trimer.

**Fig. 4 Influence of chain mobility on the thermoresponse of a SW trimer. a**, **b** Atomic configurations of the SW trimer initially (**a**), and after 1 μs long MD simulation at 450 K (**b**). **c** Time-evolution of the average distance between the molecule cores (top) and chain fluctuations (bottom). The insets are MD snapshots before and after expansion event 2 occurring at 450 K. **d** Time-averaged evolution of the total energy of the SW trimer during the MD simulation at 450 K.

rise to a giant thermal expansion the whole molecular network (an effect that is not disturbed even at step edges of the Au(111) surface).

**Generalization of interdigitated 2D supramolecular networks to large thermal expansion**. Besides unravelling the pivotal role of the alkyl chains into the giant thermal expansion of the SW assembly, we now discuss the generalization of this mechanism to interdigitated molecular systems at surfaces. Table 1 compares the lattice constants $a$ measured at 5 and 300 K and the extracted thermal expansion $\alpha$ of the SW assemblies with hexabenzocoronene (HBC) assemblies with and without alkyl chains (see

Supplementary Note 4). In a recent work[24], we studied the assembly of HBC molecules equipped with six dodecyl side chains (HBC-6C$_{12}$H$_{25}$) on Au(111) at 5 and 300 K by combined STM/AFM measurements (see Supplementary Fig. 7). From the lattice measurements, we extract of a thermal expansion of this system of about $\alpha = 520 \pm 60 \times 10^{-6}$ K$^{-1}$. While the same hexagonal ordering has been reported by Meissner et al. [34] from pristine HBC molecule (without alkyl chains), no lattice variation has been observed between 1.2 K and 300 K STM measurements, resulting in a thermal expansion of only $\alpha = 14 \times 10^{-6}$ K$^{-1}$[34,35].

Both the SW and HBC-6C$_{12}$H$_{25}$ molecular assemblies experience a similar giant thermal expansion, that we attribute to the increased fluctuations of their interdigitated long alkyl chains governing the intermolecular assemblies. The cores of these precursors are quite different accounting for the slight difference in observed thermal expansion since the HBC lies flat on the surface[24] while the SW core tends to protrude upon adsorption. In contrast to these interdigitated systems, supramolecular networks formed by pristine HBC molecules do only show a negligible expansion[34,35]. This observation again points towards the central role of the alkyl chains in the thermal expansion mechanism of interdigitated systems. Based on these observations, we believe that future works might address further the influence of the molecule cores as well as the effect of the chain lengths. We also anticipate that the structural phase transition might be also observed in such alkyl-based expansion mechanism[24]. This might enable tailoring the thermoresponse of such supramolecular systems.

## Discussion
We investigated the giant thermal expansion of a 2D supramolecular assembly composed of SW molecules on Au(111). Combining high-resolution STM/AFM images with MD calculations, we quantified a thermal expansion coefficient of $\alpha = 980 \pm 110 \times 10^{-6}$ K$^{-1}$ value that can be explained by temperature-dependent fluctuations of alkyl side chains compared to the core of the molecules. The non-covalent intermolecular interaction governed by the alkyl chain interdigitation is thus the main factor responsible for the network expansion contrarily to the molecule core found to have a marginal role in the process. This mechanism is further generalised to two-dimensional interdigitated supramolecular systems at surfaces, in contrast to their pristine counterparts (without side chains) showing no thermal expansion. Such generalization might allow the rational design of new thermoresponsive 2D materials.

## Methods
**Sample preparation**. Au(111) single crystals (Mateck GmbH) were prepared in UHV conditions by several cycles of Ar$^+$ sputtering and annealing at 750 K. As

result, atomically flat surfaces were obtained with large terraces separated by atomic steps.

**Electrospray deposition**. The electrospray deposition was performed on Au(111) samples kept at room temperature using a commercial system from Molecular-Spray. The setup was connected to preparation chambers of both low and room temperature systems. The SW molecules were dissolved in a toluene and methanol mixture (ratio 5 to 1 in volume). During the spray deposition the pressure rose up to $1 \times 10^{-6}$ mbar. Typical applied voltage were 1.5–2.0 kV with adjustment sometimes necessary during the spray to maintain stable conditions. Depositions were performed for around 30 min and were followed by a slight annealing at approx. 450 K in order to remove any solvent contaminants. We reproduced such depositions in both UHV apparatus with identical parameters that always lead to molecular coverage well-below the monolayer and non-uniform molecular densities at the macroscopic scale. This allowed us to investigate both single molecules and extended assemblies on the same sample (see Supplementary Figs. 1 and 2).

**Room temperature AFM**. Room temperature nc-AFM measurements were performed with a home-built non-contact atomic force microscope with Nanonis RC5 electronics. PPP-NCL cantilevers (Nanosensors) were used as sensors (typical resonance frequencies of $f_{1st} = $ 150 kHz and $f_{2nd} = 1$ MHz, oscillation amplitude 2–5 nm and 200–800 pm, respectively). Their preparation consisted in an annealing for 1 h at 400 K followed by a tip $Ar^+$ sputtering for 90 s at 680 eV at an $Ar^+$ pressure of $p = 3 \times 10^{-6}$ bar. Base pressure of UHV system was maintained to $2 \times 10^{-11}$ mbar during the measurements. The calibration of the ($XY$) AFM scanner at 300 K was checked by imaging the atomic surface lattice of a KBr(001) crystal.

**Low temperature AFM/STM**. Low temperature STM/AFM measurements were realized with a low-temperature microscope (Omicron Nanotechnology GmbH) operated at 5 K with Nanonis RC5 electronics and based on a tuning fork sensor in the qPlus configuration (resonance frequency f = 25 KHz, at 5 K, oscillation amplitude of $A = 50$ pm). The STM experiments were conducted in the constant-current mode. The AFM experiments were performed in constant-height mode with a CO-terminated tips [21]. The calibration of the ($XY$) scanner was checked by imaging the Au(111) surface with a CO-terminated tip.

**SW synthesis**. The SW molecules were prepared by multistep organic synthesis, through six-fold intramolecular Yamamoto coupling reaction of a carefully designed polyphenylene precursor bearing 12 bromo groups, which was obtained through a cobalt-catalysed cyclotrimerization of corresponding diarylacetylenes. The details of the synthesis and characterizations are reported in Ref. [17].

**Molecular dynamic (MD) simulation details**. MD simulations were performed using AMBER18[36] with NVIDIA GPU acceleration[37,38]. To account for long-range electrostatic interactions, periodic boundary conditions and Particle Mesh Ewald[39,40] were used with standard defaults and a real-space cutoff of 1.5 nm. Van der Waals interactions were truncated at the real space cutoff, and Lorentz-Berthelot mixing rules were used to determine the interaction parameters between different atoms. All simulations are preformed in vacuum conditions and the volume of the system was kept fixed and the temperature was adjusted by means of a Langevin thermostat [41] with a damping rate of $1 ps^{-1}$ which ensures fast thermalization with a minimal effect on the dynamics[33] of the arms and slip dynamics governing the diffusion of the SW core. The SHAKE algorithm [42] was used to constrain bonds containing hydrogen, thus allowing the use of an integration time step of 1 fs. Coordinates were saved every 1000 steps.

**Atomic level models and force fields**. An Au(111) surface composed by three atomic layers, where the positions of the atoms in the lowest layer were fixed during the MD runs using a harmonic restrain of 5 kcal mol$^{-1}$, was considered. Surfaces of two different sizes were considered, i.e. $10 \times 10$ nm$^2$ (used in single molecule simulations shown in Fig. 3 and Supplementary Fig. 3) and $16 \times 16$ nm$^2$ (used in three molecules simulations shown in Fig. 4 and Supplementary Fig. 4). SW initial structure was generated using the Avogadro software (see Supplementary Fig. 3a, b). Subsequently, the structure was relaxed via the energy minimization procedure described in the simulation protocol section. The interaction between the atoms composing the SW molecule was described using the *General Amber Force Field* (GAFF)[43–45]—including its latest refinements (version 1.8 from 2017) available in AMBER18 [36] that better reproduce the high-level ab initio and high quality experimental data of a variety of molecular properties[36]. At the intramolecular level, GAFF is able to accurately reproduce the structure and vibrational properties of a wide range of organic molecules [43–45] (including biphenyl and aliphatic chains which are the basic building blocks of the SW). Moreover, at the intermolecular level, GAFF has been routinely and successfully applied to study supramolecular polymers[45,46], mechanically interlocked molecules [47], molecular assembly processes[48,49] as well as other thermophysical properties that strongly depend on intermolecular interactions[45,50] (e.g. liquid density, heat of capacity, self-diffusivity among others). As for the gold atoms, the CHARMM-METAL force-field [51–53] was applied, which simultaneously describe the intrinsic

properties of gold, while retaining thermodynamic consistency with AMBER force field [51–53]. This force field has been extensively tested by studying the adsorption of small organic molecules (charged and uncharged and within the same chemical space as SW) against both density-functional-theory simulations as well as available experimental results[51–53]. Moreover, its suitability to describe the interaction with small organic molecules and their surface mobility has been recently validated through single molecule manipulation experiments[33,54].

**Simulations protocols**. In total six different microsecond long MD simulations, that can be grouped in two stages, were performed. The simulation details, notably the preparation stages prior to production runs, are provided in Supplementary Note 1. Here only a brief outline of each stage is provided. Stage 1: Starting from the atomic configuration obtained after 10 ns long MD simulation of the SW adsorption at 300 K to Au(111), 4 different microsecond long MD simulations were performed. Each performed at a fixed constant temperature (i.e. at 5, 300, 420 and 450 K) allows the investigation of the influence of the temperature in the diffusion of the core as well as the chain mobility. Stage 2: In order to inspect the influence of the thermally enhanced arm mobility in the molecular assembly, two different microsecond long simulations were performed consisting in an assembly of three SW molecules (as shown in Fig. 4). In this case only the limiting temperatures, i.e. 5 and 450 K, were considered. At low temperatures the apparent lack of mobility of both the core and the chains simply confirms the stability of the assembly observed experimentally. At high temperatures (T = 450 K), and by performing very long MD simulations (microsecond long) one could observe the diffusion of the molecular bodies assisted by the chain mobility. Lower values of temperature were deliberately not considered, as in those cases the timescales of the diffusion dynamics increase significantly (as there is less thermal energy available) rendering the those MD simulation unfeasible, i.e. well beyond the current state of the art microsecond long regime on such large scale systems.

## Data availability

The data that support the findings of this study are available from the corresponding authors upon reasonable request.

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

## Acknowledgements

Se.S., A.H., R.P., S.F., Z.L., T.G. and E.M. thank the Swiss National Science Foundation (SNF) and the Swiss Nanoscience Institute (SNI). J.G.V. acknowledges funding from a Marie Sklodowska-Curie Fellowship (DLV-795286) within the Horizons 2020 framework and also the computer resources, technical expertise and assistance provided by the Red Española de Supercomputación at the Power9 Supercomputer (BSC, Barcelona). E.M., R.P. and Se.S. thank the funding from the European Research Council (ERC) under the European Union's Horizon 2020 research and innovation program (ULTRADISS grant agreement No. [834402]).

## Author contributions

E.M. and A.H. conceived the experiments. Se.S and A.H. performed spray deposition. Se.S. and A.H. performed the AFM measurements at room temperature, R.P. performed the experiments at low temperature and J.G.V conducted the simulations. A.N., Y.L. and X.F. synthesized the molecules. Se.S., A.H. and J.G.V. wrote the paper with the help of R.P. Se.S., A.H., R.P., J.G.V., Y.L., S.F., Z.L., X.F., K.M., T.G., A.N. and E.M. read and discussed the paper.

## Competing interests

The authors declare no competing interests.
