## [Peer Review File · Communications Materials]

Web links to the author's journal account have been redacted from the decision letters as indicated to maintain confidentiality.

Decision letter and referee reports: first round

2nd December 2019

Dear Mr Scherb,

Thank you for submitting your manuscript, "Giant thermal expansion of a two-dimensional supramolecular network triggered by alkyl chain motion", to Communications Materials. It has now been seen by 3 referees. You will see from their comments below that while they find your work of interest, some important points are raised. We are interested in the possibility of publishing your study in Communications Materials, but would like to consider your response to these concerns in the form of a revised manuscript before we make a final decision on publication. We therefore invite you to revise and resubmit your manuscript, taking into account the points raised.

When submitting your revised manuscript, please include the following:

-A rebuttal letter with a point-by-point response to each of the referee comments and a description of changes made. Please include the complete referee report in the rebuttal letter. Please note that the rebuttal letter must be separate to the cover letter to the editors.

-A marked-up version of the manuscript with all changes to the text in red colored font. Please do not include tracked changes or comments. Please select the file type 'Revised Manuscript - Marked Up' when uploading the manuscript file to our online system.

-A clean version of the manuscript. Please select the file type 'Article File'.

-An updated <https://www.nature.com/documents/nr-editorial-policy-checklist.pdf> Editorial Policy checklist, uploaded as a 'Related Manuscript File' type. This checklist is to ensure your paper complies with all relevant editorial policies. If needed, please revise your manuscript in response to these points. Please note that this form is a dynamic 'smart pdf' and must therefore be downloaded and completed in Adobe Reader, instead of opening it in a web browser.

Your manuscript should also comply with our format requirements, which are summarized on the following checklist:

<https://www.nature.com/documents/commsmat-checklist.pdf>>Communications Materials formatting checklist. Please modify your manuscript according to this checklist.

Please use the following link to submit your revised manuscript files:

[link redacted]

We hope to receive your revised paper within three months; please let us know if you aren't able to submit it within this time so that we can discuss how best to proceed. If we don't hear from you, and the revision process takes significantly longer, we will close your file. In this event, we will still be happy to reconsider your paper at a later date, as long as nothing similar has been accepted for publication at Communications Materials or published elsewhere in the meantime.

Please do not hesitate to contact me if you have any questions or would like to discuss these revisions further. We look forward to seeing the revised manuscript and thank you for the opportunity to review your work.

Best regards,

John Plummer, PhD

Chief Editor

orcid.org/0000-0003-4824-8497

Communications Materials

Reviewers' comments:

Reviewer #1 (Remarks to the Author):

As far as I know, this paper describes the first study of the thermal expansion coefficient of a self-assembled molecular network, as revealed by UHV STM and AFM. The special feature of this molecular system is the fact that it is substituted by alkyl chains, and that alkyl chain interdigitation is at the origin of the stability of the formation of the supramolecular network. The flexibility of the alkyl chains plays a key role.

This manuscript gives a unique insight into the mechanism of such thermal expansion. In addition to the experimental data, dynamics simulations suggest that the transition between different stages are abrupt events.

The data are well presented, and details on the analysis are provided.

I support publication of this manuscript.

I have a few questions out of interest.

I suppose that at submonolayer coverage several domains are formed. Did the authors do any statistics? I suppose that at 4K all domains have the same unit cell area, and I suppose the same will hold at 300 K. If the authors were able also to measure at temperatures in between, I suppose that all domains at given temperature would show the same lattice parameters, given the fast dynamics. Is that a correct assumption?

To what extent is the expansion of the supramolecular polymer mediated by the expansion of the gold lattice?

Is the phenomenon independent of the surface coverage?

Reviewer #2 (Remarks to the Author):

In this manuscript, Scherb et al. reported a giant thermal expansion of a polyphenylene spoked wheel molecular network on Au(111) substrate using STM/AFM and molecular dynamic (MD) simulations at different temperatures. The sample is prepared using state-of-the-art electrospray deposition (ESD) technique. The giant expansion originates from the thermal mobility of the alkyl chains. The manuscript is well written and the MD simulations support their experimental results.

The findings are interesting and could potentially be published in Communications Materials, if the following concerns are properly addressed:

1. On page 4, when introducing the ESD technique, papers from some other groups should also be cited, e.g., from Klaus Kern, Richard Berndt and Giovanni Costantini groups.
2. The authors have to prove/clarify that the molecules coverages are the same at different temperatures, and they should be sub-monolayer. Otherwise, the result is also a coverage/strain dependent phenomenon, not only related to thermal response. There should be some clean surface areas on the sample at least at 5K since the molecules are more close-packed.
3. Related to question 2, in the supplementary material the authors mention that the molecules do not desorb at 450K and it is confirmed experimentally. It would be good to show the proofs like adding more images of the sample at different areas or providing some statistical results.
4. The authors claim that they used the most compact structure possible as the starting point of the MD simulations, which is 3.9 nm. However, the lattice parameter of the supramolecular network at 5K is 3.5 nm. This point should be clarified.
5. Typo, the last sentence of the third paragraph on page 4 of the supplementary material, there is no Supplementary Figure 4d.

Reviewer #3 (Remarks to the Author):

The manuscript by Schreb and coworkers describes combined experimental and theoretical studies on the thermal expansion of a supramolecular network adsorbed on gold (111) crystalline surface. The Authors performed ultra-high vacuum STM measurements of the self-assembled overlayers comprising the spoked wheel (SW) molecule equipped with six peripheral dodecyl chains. These measurements were compared with the theoretical counterparts obtained with the Molecular Dynamics simulations. The main objective of these investigations was the determination of the influence of temperature on the intrinsic parameters of the investigated systems, in particular on the molecular core-core distance which characterizes the degree of thermal expansion. The results obtained with both methods proved the exceptionally high thermal expansion coefficient of the SW-based networks, being of order of $900 \times 10^{-6}/K$. These findings are very interesting and demonstrate the thermal expansion mechanism which is based on the mobility of the side dodecyl

chains. The value of the thermal expansion coefficient reported in the manuscript is about twice larger than that measured previously for the coronene-based molecules [ref. 24] and it has been largest obtained to date for such 2D adsorbed structures. The results of the manuscript, in both theoretical and experimental aspects, are very clearly presented and they firmly support the obtained conclusions. The reported data are novel and sound, and they can stimulate the research in such fields as material engineering and surface-assisted fabrication of stimuli-responsive molecular systems. In conclusion, I think that the paper can be published in the Nature Communications Materials provided the following minor points have been addressed:

1) The Authors do not mention about the chirality of the obtained molecular assemblies. Was it possible to distinguish in the STM imaging molecular arm interdigitations having the opposite sense of rotation (especially at low temperatures when the arms of 3 neighboring molecules form strongly tightened "spirals")? If so, did the Authors observed the formation of domains of both types or some (less likely) chiral symmetry breaking was observed?

2) In the MD calculations only three SW molecules are simulated at maximum. When looking at the extended experimental overlayer comprising hundreds of molecules, this number seems rather small. I understand that, the full MD representation of the molecules eliminates the possibility of using large sets adsorbed SWs – and this is nicely explained in the SI. However, some mention of what can be the effect of surface coverage on the simulated thermal expansion coefficient should be provided in the main text. If the molecular cores can move upon increasing mobility of the side chains, then for three of them there is a lot of space to expand in the particular simulation box used in the calculations. What if the surface SW density was higher as in the real domain? Do the Authors think that this effect would seriously affect the measured values of the thermal expansion coefficient.

3) What will happen in the simulations with the 3-molecular assembly when the calculations are run beyond 1000 ns at 450 K? In other words, is the last plateau observed in Fig. 4c is truly the last one. Is it sure 100% that the molecules will eventually not separate. If the inset after 1 micro_s is the final one then it would be useful to use some kind of dynamic time counter in the MD movies, so that the reader can easily see the stable state at 1000 micro_s and also to trace the distinct steps leading to the thermal expansion.

4) A similar question refers to the temperatures higher than 450 K. What happens then in the experiment and in the simulations? Is there a phase transition in the adsorbed overlayer? In MD simulations does the cluster disassembly rapidly or partial desorption (of dodecyl arms) can be observed?

Reviewer #1:

As far as I know, this paper describes the first study of the thermal expansion coefficient of a self-assembled molecular network, as revealed by UHV STM and AFM. The special feature of this molecular system is the fact that it is substituted by alkyl chains, and that alkyl chain interdigitation is at the origin of the stability of the formation of the supramolecular network. The flexibility of the alkyl chains plays a key role. This manuscript gives a unique insight into the mechanism of such thermal expansion. In addition to the experimental data, dynamics simulations suggest that the transition between different stages are abrupt events. The data are well presented, and details on the analysis are provided. I support publication of this manuscript. I have a few questions out of interest:

We thank the reviewer for the positive comments and deeply appreciate her/his interest in our findings.

I suppose that at submonolayer coverage several domains are formed. Did the authors do any statistics? I suppose that at 4K all domains have the same unit cell area, and I suppose the same will hold at 300 K. If the authors were able also to measure at temperatures in between, I suppose that all domains at given temperature would show the same lattice parameters, given the fast dynamics. Is that a correct assumption?

The reviewer assumption is correct. Upon spray-deposition (in the sub-monolayer range), several aggregate sizes are found on the surface varying from single molecules up to large extended assemblies. The measured lattice parameters of large assemblies are always identical for a given

temperature (5 K or 300 K), whereas the spacing between only few interacting molecules can be much larger. In our manuscript, we only refer to the lattice parameters of large domains.

As mentioned by the referee, the influence of the domain size is an important aspect of the expansion mechanism but would require an in-depth statistical analysis in real-space or the use of diffraction techniques. Even though it would certainly be an interesting study, this is slightly beyond the scope of the present work. Regarding intermediate temperatures, we could not perform such experiments during our study since we used two microscopes that operate at 5 K and 300 K, respectively. We currently work on experimental improvements to explore these interesting aspects, that will be discussed in a future work.

To clarify the referee comments, we added modified the manuscript as follow:

> Modification #1, page 5 of the main manuscript

“Thermal response of the SW assembly. Upon spray-deposition of the spoked-wheel molecules in the sub-monolayer range, we observed different aggregates on the surface varying from single molecules up to extended assemblies (see Supplementary Figure 1b and 2). To investigate the effect of temperature on the SW assembly, extended close-packed networks (at least 200 nm in width and length) chosen to be of similar size to avoid influences of the domain size on the lattice parameter were investigated. Figures 2a and b show STM and AFM images of the networks acquired at 5 K and 300 K, respectively.”

To what extent is the expansion of the supramolecular polymer mediated by the expansion of the gold lattice?

We thank the referee for this interesting point that is not discussed in the manuscript. The expansion of the gold lattice over the investigated temperature ranges is about $14 \times 10^{-6} \text{ K}^{-1}$, i.e. nearly two orders of magnitude smaller than our supramolecular network. Therefore, we expect the gold expansion to have a rather marginal role in the measured network sizes.

We commented on this in the manuscript as follows:

> Modification #2, Main manuscript page 6

“In conclusion, a thermal expansion coefficient of the SW on gold of $\alpha = 980 \pm 110 \times 10^{-6} \text{ K}^{-1}$ is determined, which is two orders of magnitude larger than the expansion of the Au(111) of about $14 \times 10^{-6} \text{ K}^{-1}$.”

Is the phenomenon independent of the surface coverage?

In this work we analysed only sub-monolayer coverage. Therefore, based on the available data we cannot answer this. We, anyhow, expect an influence at coverages close to monolayers due to strain, but we expect the mechanism of expansion to hold true for different levels of surface coverage.

Reviewer #2

In this manuscript, Scherb et al. reported a giant thermal expansion of a polyphenylene spoked wheel molecular network on Au(111) substrate using STM/AFM and molecular dynamic (MD) simulations at different temperatures. The sample is prepared using state-of-the-art

electrospray deposition (ESD) technique. The giant expansion originates from the thermal mobility of the alkyl chains. The manuscript is well written and the MD simulations support their experimental results. The findings are interesting and could potentially be published in Communications Materials, if the following concerns are properly addressed:

We thank the reviewer for his/her well thought out review and the important concerns raised. We, also, appreciate his/her recognition of the interest of this work.

1. On page 4, when introducing the ESD technique, papers from some other groups should also be cited, e.g., from Klaus Kern, Richard Berndt and Giovanni Costantini groups.

We thank referee for bringing this to our attention. We added citations of Richardt Berndt, Giovanni Costantini and Klaus Kern groups in the introduction, where we first discuss the ESD technique. We added the following citations:

- Hauptmann, N., Hamann, C., Tang, H. & Berndt, R. Switching and charging of a ruthenium dye on Ag(111). *Phys. Chem. Chem. Phys.* **15**, 10326-10330 (2013)

- Rinke, G. *et al.* Soft-landing electrospray ion beam deposition of sensitive oligoynes on surfaces in vacuum. *Int. J. Mass Spectrom.* **377**, 228-234 (2015).

- D. A. Warr *et al.* Sequencing conjugated polymers by eye. *Science Advances*, **4**, 9543 (2018).

They appear as reference 29,30 and 32 in the introduction of the manuscript as follow:

> Modification #1, page 3/4 of the main manuscript

“In UHV conditions, the electrospray deposition (ESD) technique allows the introduction of large molecules. Therefore, it is possible to study in UHV conditions the adsorption properties of molecules and their assemblies that mimic established assemblies at the liquid-solid interface.^{24,27-33} ...To safely deposit the SW precursors onto Au(111), we used an electrospray deposition (ESD) device attached to the UHV setup^{24, 27-33}”

2. The authors have to prove/clarify that the molecules coverages are the same at different temperatures, and they should be sub-monolayer. Otherwise, the result is also a coverage/strain dependent phenomenon, not only related to thermal response. There should be some clean surface areas on the sample at least at 5K since the molecules are more close-packed.

The referee raises an important point here that is loosely described in our work. Upon spray-deposition, we never reached a full monolayer coverage at the surface. Furthermore, we observed an important variation of molecule density at different macroscopic areas, that allowed us to investigate single molecules and densely packed assemblies on the same samples. This is illustrated by the STM overviews obtained on different regions and show below.

Between our UHV systems, the deposition parameters are very similar. We thus obtained similar situations with various molecule densities for the study at 5 K and 300 K, respectively. To clarify this point, we added these STM/AFM images to the Supplementary Information as Fig. S1.

> Modification #2, page 2 of the Supplementary Information

Supplementary figure 1: **Isolated spoked wheel molecules on Au(111)**. (a) Topography STM image at 5~K of SW molecules on Au(111) surface after spray deposition onto a surface maintained at room temperature, prior to annealing. (b) Topography STM image of similar density area after annealing at 450~K, observed after microscale tip displacement from Fig. 2 a. Isolated SW molecule on Au(111) surface and step edges at 5 K are visible as pointed with red arrow. Scale bar is 50~nm.

> **Modification #3, page 3 of the Supplementary Information**
 “Supplementary Figure 2: Large scale assemblies on Au(111)”

Supplementary Figure 2: **Large scale assemblies on Au(111)**. (a) Topography STM image of SW molecule high density area on Au(111) surface at 5~K after spray deposition and prior to annealing. (b) Topography STM image of SW molecule assemblies on Au(111) surface at 5~K after annealing at 450~K. (c) Topography AFM image at 300~K of SW molecule assemblies on Au(111) surface after annealing at 450~K. Scale bars are 50~nm.

We also commented on this point in the Methods as follows:

> **Modification #2, Methods of the manuscript**

“We reproduced such depositions in both UHV apparatus with identical parameters that always lead to molecular coverage well-below the monolayer and non-uniform molecular densities at the macroscopic scale. This allowed us to investigate both single molecules and extended assemblies on the same sample (see Supplementary Figures 1 and 2).”

3. Related to question 2, in the supplementary material the authors mention that the molecules do not desorb at 450K and it is confirmed experimentally. It would be good to show the proofs like adding more images of the sample at different areas or providing some statistical results.

To address the referee comments, we added STM overviews of different macroscopic areas in Supplementary Figure 1. We systematically observed various molecule densities at the macroscopic scale, well-below the monolayer, varying from the singles molecules up to large assemblies on the same sample. We also commented in the main manuscript on that as follows:

> Modification #5, page 4 of the main manuscript

“To safely deposit the SW precursors onto Au(111), we used an electrospray deposition (ESD) device attached to the UHV setup. [24,28-30] The spray-deposition was always performed on the Au(111) substrates kept at room temperature and followed by an annealing step at 450 K. As shown in large-scale STM measurements (Supplementary Figure 1), single SW molecules are observed intact on the surface indicating the low diffusion of the molecule on the gold surface and the absence of any degradation.”

4. The authors claim that they used the most compact structure possible as the starting point of the MD simulations, which is 3.9 nm. However, the lattice parameter of the supramolecular network at 5K is 3.5 nm. This point should be clarified.

We fully agree with the referee and sincerely appreciate his/her thorough revision. This point is now clarified in the revised version of both the manuscript and Supplementary Material as shown below.

> Modification #6, page 8 of the main manuscript

“... is numerically obtained by comparing MD simulations of a SW trimer at different temperatures. In the interest of the computational feasibility, given the sheer size and required simulation time of large assemblies, we considered a trimer whose core-core distance matched an equivalent assembly observed at 5K (see Supplementary Figure 4a). Further details on the simulation protocol are provided in Supplementary Note 1. Interestingly, small molecular assemblies displayed a core-core distance slightly bigger than larger assemblies, an effect possibly resulting from the domain size. The simulations of this molecular assembly ... The associated thermal expansion coefficient is thus $600 \pm 120 \times 10^{-6} \text{ K}^{-1}$ in agreement with the experimental value.”

> Modification #7, page 4/5 of the Supplementary Information

“The thermal stability of the SW molecular assembly was evaluated by using three SW, packed in a conformation that matches the inter-molecular distance measured in the low temperature experiments of small molecular assemblies. Such a molecule trimer was imaged by LT STM, as visible in Supplementary Figure 4a, and gives an average distance between molecules of 3.8 to 4.2 nm (which is slightly larger than the average distance in large molecular assemblies shown in Fig. 2 – $d=3.5 \text{ nm}$).”

> Modification #8, page 5/6 of the Supplementary Information

“Firstly, we had to build an assembly that matches the compact structure of small molecular assemblies – which best describe our simulation setup – such as the one shown in Supplementary Figure 4a. The reason for this ... Note that as the molecules diffuse inwards in order to give rise to a more compact structure they also form an energetically more favourable configuration as shown in Supplementary Figure 4d. Interestingly, an indirect evidence for this effect can be found in our experiments as small molecular assemblies (see Supplementary Figure 4a) have a bigger core-core distance with respect to the large molecular assemblies (see Figure 2).”

> Modification #9, page 7 of the Supplementary Information

“For this reason, a compact structure mimicking a small molecular assembly (shown in Supplementary Figure 4a) was chosen as starting point and then heated up. The heating process ...”

5. Typo, the last sentence of the third paragraph on page 4 of the supplementary material, there is no Supplementary Figure 4d.

We thank the reviewer for finding this oversight. Since we added an additional figure to the Supplementary Information the reference does now point to the correct figure. We revised the manuscript and SI accordingly.

Reviewer #3

The manuscript by Schreb and coworkers describes combined experimental and theoretical studies on the thermal expansion of a supramolecular network adsorbed on gold (111) crystalline surface. The Authors performed ultra-high vacuum STM measurements of the self-assembled overlayers comprising the spoked wheel (SW) molecule equipped with six peripheral dodecyl chains. These measurements were compared with the theoretical counterparts obtained with the Molecular Dynamics simulations. The main objective of these investigations was the determination of the influence of temperature on the intrinsic parameters of the investigated systems, in particular on the molecular core-core distance which characterizes the degree of thermal expansion. The results obtained with both methods proved the exceptionally high thermal expansion coefficient of the SW-based networks, being of order of $900 \times 10^{-6}/K$. These findings are very interesting and demonstrate the thermal expansion mechanism which is based on the mobility of the side dodecyl chains. The value of the thermal expansion coefficient reported in the manuscript is about twice larger than that measured previously for the coronene-based molecules [ref. 24] and it has been largest obtained to date for such 2D adsorbed structures. The results of the manuscript, in both theoretical and experimental aspects, are very clearly presented and they firmly support the obtained conclusions. The reported data are novel and sound, and they can stimulate the research in such fields as material engineering and surface-assisted fabrication of stimuli-responsive molecular systems. In conclusion, I think that the paper can be published in the Nature Communications Materials provided the following minor points have been addressed:

We are very grateful for thorough revision and constructive criticism provided by the referee. Furthermore, we appreciate that the referee recognized the interest of the current work. In what follows we carefully address all the issues raised.

1) The Authors do not mention about the chirality of the obtained molecular assemblies. Was it possible to distinguish in the STM imaging molecular arm interdigitations having the opposite sense of rotation (especially at low temperatures when the arms of 3 neighboring molecules form strongly tightened “spirals”)? If so, did the Authors observed the formation of domains of both types or some (less likely) chiral symmetry breaking was observed?

The reviewer raises an interesting question, which we cannot conclude on based on the present study data. In a previous work [ref. 24], we indeed reported the compression of the coronene-based molecules accompanied by a symmetry breaking and the emergence of a chirality character in molecular domains. We assume that this effect might be also encountered in the SW systems although we do not currently have data to prove it.

We commented in the manuscript regarding this as follows:

> Modification #10, page 12 of the manuscript

“Based on these observations, we believe that future works might further address the influence of the molecule cores as well as the effect of the chain lengths. We also anticipate that structural phase transition might be also observed in such alkyl-based expansion mechanism [24].”

2) In the MD calculations only three SW molecules are simulated at maximum. When looking at the extended experimental overlayer comprising hundreds of molecules, this number seems rather small. I understand that, the full MD representation of the molecules eliminates the possibility of using large sets adsorbed SWs – and this is nicely explained in the SI. However, some mention of what can be the effect of surface coverage on the simulated thermal expansion coefficient should be provided in the main text. If the molecular cores can move upon increasing mobility of the side chains, then for three of them there is a lot of space to expand in the particular simulation box used in the calculations. What if the surface SW density was higher as in the real domain? Do the Authors think that this effect would seriously affect the measured values of the thermal expansion coefficient.

We agree with the referee and we have now discussed the possible effect of surface coverage also in the main text of the manuscript and further expanded on in the Supplementary Information.

> Modification #11, page 10/11 of the main manuscript

“The high lattice expansion may be understood as a conjunction of three properties of the dodecyl chains: ... and excluding chain fluctuations out of the surface plane.

Although the finite size of our simulations might affect the chain fluctuations and the inter-molecular interaction, our simulations clearly support an abnormal thermal expansion coefficient for alkyl decorated molecules. Interestingly, small molecular assemblies observed experimentally show a larger core-core distance (see Supplementary Figure 4a) suggesting that the size of the molecular domains affect the intermolecular spacings. This might explain the smaller thermal expansion coefficient obtained in our simulations. At last, the large thermal expansion coefficient obtained in the experiments seems to support the idea that although in smaller assemblies the cores possess an initial higher mobility, in the larger assemblies the long experimental time scale allows enough time for the chain fluctuations equilibrate giving rise to a giant thermal expansion the whole molecular network (an effect that is not disturbed even at step edges of the Au(111) surface).”

> Modification #12, page 12 of the main manuscript

“Table 1: Thermal expansion coefficient of interdigitated 2D molecular systems ... supramolecular assemblies of spoked wheel molecules (SW), pristine hexabenzocoronene molecules (HBC) and hexabenzocoronene molecules functionalized with alkyl chains (HBC-6C₁₂H₂₅). SW, HBC-6C₁₂H₂₅ and HBC are measured on large domains, SW calculated on a trimer.”

> Modification #13, page 5 of the Supplementary Information

“Additionally, larger assemblies will certainly take longer to equilibrate as a result of a larger phase space and the required concerted motion of the assembly as a whole, which at the experiments time scale (minutes) would not play an important role.”

3) What will happen in the simulations with the 3-molecular assembly when the calculations are run beyond 1000 ns at 450 K? In other words, is the last plateau observed in Fig. 4c is truly the last one. Is it sure 100% that the molecules will eventually not separate. If the inset after 1 micro_s is the final one then it would be useful to use some kind of dynamic time counter in the

MD movies, so that the reader can easily see the stable state at 1000 micro_s and also to trace the distinct steps leading to the thermal expansion.

We sincerely appreciate the referee suggestion and we have included a time counter in all our animations.

Although it is impossible to ascertain that a certain configuration is a true equilibrium one, the results presented soundly demonstrate the importance of anharmonic vibrations of the alkyl chains on the gigantic thermal expansion process observed. The stark contrast between the single molecule (Fig.3) and trimer (Fig.4) 1 μ s long dynamics at 450K shows that the molecule alone is unable to diffuse (see Fig.3b) due to its sheer size (and consequent large surface interaction). Therefore, diffusion and subsequent thermal expansion must be assisted by the anharmonic vibrations of the alkyl chains. As a result, it is sensible to assume that the molecules will not be able to fully separate, as they are unable to diffuse by themselves.

Concerning the stability of the last plateau, it is important to notice that the time between expansion events increases significantly (first expansion occurs after 50ns, the second 150ns after the first, the third and last expansion takes over 400ns). This shows that the slip/diffusion events become less and less frequent (or equivalently with higher and higher activation energies) as the molecules move apart. Considering this and that the alkyl chains are almost straight at the end of 1 μ s long dynamics (see Fig. 4b), thus having enough space to fluctuate (see Fig.4c) and not requiring to push neighbouring molecular cores to fluctuate, it is reasonable to consider that this final configuration is an equilibrium one. At last it is interesting to compare the core-core distance of an our equilibrated trimer (4.9 nm) to the core-core distance of three SW with the alkyl chains fully stretched (4.8nm – see Supplementary Figure 3). Once the chains are fully stretched the expansion mechanism of the core (as shown in the inset of Fig.4c) is suppressed. All in all, our results seem to support that this is an equilibrium configuration but even if it was not the case, the mechanism giving rise to the reported giant expansion would still hold and perhaps result in slightly larger TEC in better agreement with the experiments.

> Modification #14, Supplementary Movies 1-5

We have included a time counter in all our animations.

4) A similar question refers to the temperatures higher than 450 K. What happens then in the experiment and in the simulations? Is there a phase transition in the adsorbed overlayer? In MD simulations does the cluster disassembly rapidly or partial desorption (of dodecyl arms) can be observed?

We appreciate that the referee brought this to our attention as indeed it required further clarification. This has been mended in the new version of the manuscript (see below). For the convenience of the referee we include in the response an image of the sample after annealing to T>500 K which we did not felt necessary to include in the Supplementary Material.

> Modification #15, page 12 of the Supplementary Information

“Molecular Assembly stability at high temperatures (>500 K)

Contrary to covalently bonded materials, molecular assemblies are held together by much weaker interactions, and are therefore prone to a swift degradation at relatively lower temperatures. In fact, in our experiments we observed that after annealing $T > 500$ K the molecules either desorbed from the surface or decomposed giving rise to small fragments on the surface. On the basis of this result, and given that our interest lays on understanding the thermoresponse of a molecular assembly we found unnecessary to complement these findings with additional simulations.”

Figure 2: Topography after annealing at 500 K. AFM at room temperature.

Editorial Guidelines:

> **Modification #16, page 4 of the main manuscript**

We have adapted the colour code in Figure 1a for colour-blind readers.

> **Modification #17, page 2 of the main manuscript**

“... By comparing high-resolution scanning probe microscopy and molecular dynamics simulations obtained at 5 and 300 K we determine the thermal expansion coefficient ...”

> **Modification #18, main manuscript**

We removed punctuation from the subheadings.

Decision letter and referee reports: second round

18th December 2019

Dear Mr Scherb,

Your manuscript titled "Giant thermal expansion of a two-dimensional supramolecular network triggered by alkyl chain motion" has now been seen again by the three referees, whose comments appear below. In light of their advice I am delighted to say that we are happy, in principle, to publish a suitably revised version in Communications Materials under the open access CC BY license (Creative Commons Attribution v4.0 International License).

We therefore invite you to edit your manuscript to comply with our format requirements and to maximise the accessibility and therefore the impact of your work.

EDITORIAL REQUESTS:

* Your manuscript should comply with our policies and format requirements, detailed in our checklist for authors at:

<https://www.nature.com/documents/commsmat-checklist.pdf>

* The abstract should be edited so that the first one or two sentences introduce the topic and motivation for the present work. The next sentence, that describes what was done in the present paper, should begin with "Here we...". Please note that the abstract should not exceed 160 words. The alpha symbol should also be removed from the abstract as it is only used once.

* Please remove italic font from "thermal expansion" on the first line of the Introduction.

* A "Results" heading should be introduced just before the "Single spoked wheel..." sub-heading. A Discussion heading is also needed; this could replace the current "Conclusions" heading as we do not have "Conclusions" headings in our papers.

* For the "SW synthesis" section of the Methods, please give some information here and then refer the reader to ref. 17 for full details.

* Please upload the descriptions of the Supplementary Movies in a separate file.

* Communications Materials uses a transparent peer review system, where by we are publishing the reviewer comments to the authors, author rebuttal letters and journal decision letters of our research articles online as a supplementary peer review file. Please let us know in the cover letter when submitting the final version of your manuscript if you wish to opt in or opt out of transparent peer review. If you are concerned about the release of confidential data, we do permit redactions in the interest of confidentiality. If you would like to remove such information from these files, then please let us know specifically what information you would like to have removed. Please note that we cannot incorporate redactions for other reasons. Reviewer names will be published in the peer review files if the reviewer comments to the authors are signed by the reviewer, or if reviewers explicitly agree to release their name.

* Data availability statements and data citations policy: All Communications Materials manuscripts must include a section titled "Data Availability" at the end of the Methods section or main text (if no Methods). More information on this policy, and a list of examples, is available at <http://www.nature.com/authors/policies/data/data-availability-statements-data-citations.pdf>.

- Accession codes for deposited data
- Other unique identifiers (such as DOIs and hyperlinks for any other datasets)
- At a minimum, a statement confirming that all relevant data are available from the authors
- If applicable, a statement regarding data available with restrictions
- If a dataset has a Digital Object Identifier (DOI) as its unique identifier, we strongly encourage including this in the Reference list and citing the dataset in the Data Availability Statement.

DATA SOURCES: We strongly encourage authors to deposit all new data associated with the paper in a persistent repository where they can be freely and enduringly accessed. We recommend submitting the data to discipline-specific, community-recognized repositories, where possible and a list of recommended repositories is provided at <http://www.nature.com/sdata/policies/repositories>.

If a community resource is unavailable, data can be submitted to generalist repositories such as [figshare](https://figshare.com/) or [Dryad Digital Repository](http://datadryad.org/). Please provide a unique identifier for the data (for example a DOI or a permanent URL) in the data availability statement, if possible. If the repository does not provide identifiers, we encourage authors to supply the search terms that will return the data. For data that have been obtained from publically available sources, please provide a URL and the specific data product name in the data availability statement. Data with a DOI should be further cited in the methods reference section.

* Please check whether your manuscript contains third-party images, such as figures from the literature, stock photos, clip art or commercial satellite and map data. We strongly discourage the use or adaptation of previously published images, but if this is unavoidable, please request the necessary rights documentation to re-use such material from the relevant copyright holders and return this to us when you submit your revised manuscript.

* We are committed to ensuring clarity and avoiding ambiguity in the mathematics in our papers. Consequently, please carefully check the mathematical terms throughout your manuscript (including labels on figures and figure captions) to ensure that it conforms strictly to the following guidelines. In mathematical terms, scalar variables (e.g. x , V , χ) and constants (e.g. n , \hbar , e) should be typeset in italics, and vectors (such as r , the wavevector k , or the magnetic field vector B) should be typeset in bold without italics. In contrast, subscripts and superscripts should only be italicized if they too are variables or constants. Those that are labels (such as the 'c' in the critical temperature, T_c , the 'F' in the Fermi energy, E_F , or the 'crit' in the critical current, I_{crit}) should be typeset in roman. Please also ensure the same convention is followed in figure labels, axes, and such. Additionally, to avoid doubt, unit dimensions should be expressed using negative integers (e.g. $\text{kg m}^{-1} \text{s}^{-2}$ not kg/ms^2) or the word 'per'.

* Your paper will be accompanied by a two-sentence editor's summary, of between 250-300 characters, when it is published on our homepage. Could you please approve the draft summary below or provide us with a suitably edited version.

"The intrinsic flexibility of molecules opens the door to unusual physical properties. Now, a large thermal expansion coefficient of $980 \pm 110 \times 10^{-6} \text{ K}^{-1}$ is observed by scanning probe microscopy in a supramolecular network on a gold surface."

OPEN ACCESS:

Communications Materials is a fully open access journal. Articles are made freely accessible on publication under a [CC BY](http://creativecommons.org/licenses/by/4.0) license (Creative Commons Attribution 4.0 International License). This license allows maximum dissemination and re-use of open access materials and is preferred by many research

funding bodies.

For further information about article processing charges, open access funding, and advice and support from Nature Research, please visit https://www.nature.com/commsmat/about/open-access

SUBMISSION INFORMATION:

In order to accept your paper, we require the following:

- * A cover letter describing your response to our editorial requests.
- * The final version of your text as a Word or TeX/LaTeX file, with any tables prepared using the Table menu in Word or the table environment in TeX/LaTeX and using the 'track changes' feature in Word.
- * Production-quality versions of all figures, supplied as separate files. Figures divided into parts should be labelled with a lowercase bold a, b, and so on. To ensure the swift processing of your paper please provide the highest quality, vector format, versions of your images (.ai, .eps, .psd) where available. Text and labelling should be in a separate layer to enable editing during the production process. If vector files are not available then please supply the figures in which ever format they were compiled in and not saved as flat .jpeg or .TIFF files. Any chemical structures or schemes contained within figures should additionally be supplied as separate ChemDraw (.cdx) files. If your artwork contains any photographic images, please ensure these are at least 300 dpi.
- * The final version of the Supplementary Information (figures, tables, notes etc) in one PDF file. Please submit movies, audio files and data sets as separate files. See https://www.nature.com/commsmat/submit/submission-guidelines#supplementary-info for acceptable file formats/sizes.
- ** Please note that Supplementary Information cannot be changed after the paper has been accepted **
- * An updated Editorial Policy checklist, uploaded as a 'Related Manuscript File' type. This checklist is to ensure your paper complies with all relevant editorial policies. Please note that this form is a dynamic 'smart pdf' and must therefore be downloaded and completed in Adobe Reader, instead of opening it in a web browser.
- * If you wish, an interesting image (but not an illustration or schematic) for consideration as the banner image on our homepage. The file should be 1400x400 pixels in RGB format and should be uploaded as 'Related Manuscript File'. In addition to our home page, we may also use this image (with credit) in other journal-specific promotional material.

At acceptance, the corresponding author will be required to complete an Open Access Licence to Publish on behalf of all authors, declare that all required third party permissions have been obtained and provide billing information in order to pay the article-processing charge (APC) via credit card or invoice.

Please note that your paper cannot be sent for typesetting to our production team until we have received these pieces of information; therefore, please ensure that you have this information ready when submitting the final version of your manuscript.

[link redacted]

We hope to hear from you within two weeks; please let us know if the process may take longer.

Best regards,

John Plummer, PhD
Chief Editor
orcid.org/0000-0003-4824-8497
Communications Materials

P.S. To help the scientific community achieve unambiguous attribution of all scholarly contributions, Communications Materials encourages all authors to create and link an ORCID identifier to their account. Please ensure that all co-authors are aware that they can add their ORCIDs to their accounts, so that it will display on this paper. If they so wish, they must do so before the paper is formally accepted. It will not be possible to add ORCIDs post-acceptance, e.g. at proof. To add an ORCID please follow these instructions:

1. From the home page of the <https://mts-commsmat.nature.com/cgi-bin/main.plex> click on 'Modify my Springer Nature account' under 'General tasks'.
2. In the 'Personal profile' tab, click on 'ORCID Create/link an Open Researcher Contributor ID (ORCID)'. This will re-direct you to the ORCID website.
- 3a. If you already have an ORCID account, enter your ORCID email and password and click on 'Authorize' to link your ORCID with your account on the MTS.
- 3b. If you don't yet have an ORCID account, you can easily create one by providing the required information and then clicking on 'Authorize'. This will link your newly created ORCID with your account on the MTS.

REVIEWERS' COMMENTS:

Reviewer #1 (Remarks to the Author):

I'm happy with the replies of the authors and the changes made to the manuscript, and I recommend publication.

Reviewer #2 (Remarks to the Author):

The authors have taken into account my previous concerns and revised the manuscript accordingly. Now I recommend the publication of this manuscript.

Reviewer #3 (Remarks to the Author):

The Authors addressed in detail all of my remarks. Their explanations are fully convincing. I am now sure that the manuscript in its new form can be published in the Nature Communication Materials journal.